# The Desire to Better Understand Older Adults with Solid Tumors to Improve Management: Assessment and Guided Interventions—The French PACA EST Cohort Experience

**DOI:** 10.3390/cancers11020192

**Published:** 2019-02-07

**Authors:** Rabia Boulahssass, Sebastien Gonfrier, Noémie Champigny, Sandra Lassalle, Eric François, Paul Hofman, Olivier Guerin

**Affiliations:** 1Geriatric Coordination Unit for Geriatric Oncology (UCOG) PACA Est CHU de Nice, 06000 Nice, France; gonfrier.s@chu-nice.fr (S.G.); champigny.n@chu-nice.fr (N.C.); guerin.o@chu-nice.fr (O.G.); 2FHU OncoAge, 06000 Nice, France; lassalle.s@chu-nice.fr (S.L.); hofman.p@chu-nice.fr (P.H.); 3University Côte d’Azur, 06000 Nice, France; 4Laboratory of Clinical and Experimental Pathology, Pasteur Hospital, 06000 Nice, France; 5Hospital-related Biobank (BB-0033-00025), 06000 Nice, France; 6Department of Medical Oncology, Lacassagne Center, 06000 Nice, France; eric.francois@nice.unicancer.fr

**Keywords:** cancer, older adults, geriatric assessment, geriatric interventions

## Abstract

Todays challenge in geriatric oncology is to screen patients who need geriatric follow-up. The main goal of this study was to analyze factors that identify patients, in a large cohort of patients with solid tumors, who need more geriatric interventions and therefore specific follow-up. Between April 2012 and May 2018, 3530 consecutive patients were enrolled in the PACA EST cohort (France). A total of 3140 patients were finally enrolled in the study. A Comprehensive Geriatric Assessment (CGA) was performed at baseline. We analyzed the associations between factors at baseline (geriatric and oncologic factors) and the need to perform more than three geriatric interventions. The mean age of the population was 82 years old with 59% of patients aged older than 80 years old. A total of 8819 geriatric interventions were implemented for the 3140 patients. The percentage of patients with three or more geriatric interventions represented 31.8% (n = 999) of the population. In multivariate analyses, a Mini Nutritional assessment (MNA) <17, an MNA ≤23·5 and ≥17, a performans status (PS) >2, a dependence on Instrumental Activities of Daily Living (IADL), a Geriatric Depression Scale (GDS) ≥5, a Mini Mental State Examination (MMSE) <24, and a Screening tool G8 ≤14 were independent risk factors associated with more geriatric interventions. Factors associated with more geriatric interventions could assist practitioners in selecting patients for specific geriatric follow-up.

## 1. Introduction

Cancer is significantly associated with aging, and life expectancy is increasing in France and worldwide [1]. Thus, the proportion of older adults with cancer is rising [2]. Despite this “demographic tsunami,” elderly patients are under-represented in clinical studies [3]. The population of the French Riviera is one the oldest populations in France; a quarter of the population is over 60 years old, and the proportion of the population over 80 years old has increased by nearly 40% within the past decade [4]. In this context, this region is a “living laboratory” for elderly patients. In 2011, we set up the geriatric coordination unit for geriatric oncology (UCOG PACA EST) with the support of the French National Cancer Institute (INCA). It aims to upgrade care, research, and teaching in geriatric oncology field in Southeast France. To study this population, we decided to create a database called the “PACA EST Cohort” by developing a strong partnership between the Lacassagne Cancer Center (Nice, France) and the geriatric department of the University Hospital of Nice. In 2016, to participate in further collaborative research, the UCOG PACA EST team joined the Hospital Federation for research, OncoAge, a consortium of skills of a high level from several fields in health care, research, and education dedicated to cancer in the elderly. 

In a clinical routine, the International Society of Geriatric Oncology (SIOG) and the American Society of Clinical Oncology (ASCO) recommend performing a comprehensive geriatric assessment (CGA) due to the substantial heterogeneity among elderly patients [5,6]. A CGA as defined by Rubenstein is a “multidimensional interdisciplinary diagnostic process focused on determining a frail elderly person’s medical, psychological, and functional capability in order to develop coordinated and integrated plan for treatment” [7]. The CGA is time-consuming, but specific tools for frailty screening are available to detect patients who really need to perform a complete CGA [8,9,10]. During the past decade, the partnership between geriatricians and oncologists has improved patient care by profiling the level of patient frailty with this process. Therefore, the CGA has been shown to predict outcomes (chemotherapy toxicity, life expectancy) to help make therapeutic decisions but also provide the best interventions [11,12]. Previous studies have described adherence of geriatric assessment recommendations [13,14] as well as guidelines for practical assessment and management of older patients receiving chemotherapy [6]. Now, the challenge in geriatric oncology is to screen patients who require geriatric follow-up with specific guided interventions. The main purpose is to determine which patients need to have repeated geriatric assessment in the follow-up. Therefore, the goal of this study was to analyze a large cohort of patients with solid tumors, for factors that provide a profile of the phenotype of patients who need more geriatric interventions and therefore specific follow-up.

## 2. Results

### 2.1. Patient Characteristics

The mean age of the population was 81.9 years old (range 70–102) with 59% of patients aged older than 80 years old. Fifty-five percent were women and 33% had a metastatic status. The most common cancers observed in the cohort were breast cancers (n = 548/17.5%), colorectal cancer (n = 527/16.7%), and lung cancers (n = 356/11.3%) (Table 1).

### 2.2. Geriatric Assessment Model

Only 13.5% of patients had a G8 >14, which allows practitioners to not perform a full CGA in clinical routines. 

In the whole cohort, 16% felt homebound, 48.6% had dependence on ADL, and 16% were malnourished according to the MNA. Table 2 shows a description of the domains that were explored in the standardized CGA at baseline.

### 2.3. Treatments Proposed and Influence of the CGA

Patients were referred to a geriatrician; 47.7% were referred for treatment with chemotherapy or a combined treatment, 28% for surgery, 10.2% for radiotherapy, 8.3% for best supportive care, and 6% for other treatments. In 22% of patients, the CGA modified the therapeutic decision.

### 2.4. Geriatric Interventions

#### 2.4.1. Description

A total of 8819 geriatric interventions were implemented for the 3140 patients. On average, fit patients benefited from 1.5 geriatric interventions, patients classified as “Balducci 2” from 2.4 interventions, and frail patients from 3.3 interventions. In the whole cohort, the medium number of interventions per patient was 2.8. Vulnerable and frail patients had significantly more geriatric interventions (*p* < 0.0001). The guided geriatric interventions are listed in Table 3. 

#### 2.4.2. Factors Associated with an Increased Need of Geriatric Interventions

Patients with three or more geriatric interventions represented 31.8% (n = 999) of the population. Univariate significant factors associated with an increased need of geriatric interventions are listed in Table 4. In multivariate analyses, an MNA <17, an MNA ≤23.5 and ≥17, a PS >2, a dependence on IADL, a GDS ≥5, an MMSE ≤24, and a G8 ≤14 are independent risk factors associated with this requirement (Table 5).

## 3. Discussion

### 3.1. The Challenge in Geriatric Oncology Is to Screen Patients for Follow-Up

The first step in geriatric oncology is to screen patients who need a complete geriatric assessment. A number of tools are available and recommended for screening [8,9,10]. Thus, practitioners can propose a comprehensive assessment and elaborate recommendations according to the deficits observed [6]. In 2013, Kenis et al. [15] demonstrated that screening and CGA are feasible in clinical practice and detected unknown geriatric problems in 51% of cases. This study also showed that oncologists were aware of the geriatric assessment results only in 2/3 of the patients, and recommendations were planned in only 25%. This cohort study highlights the difficulty of implementing geriatric interventions and of the necessity of follow-up. Recent studies have listed the types of interventions and their implementation, but they did not analyze the factors that can lead to more interventions [13,14]. Baitar et al. [16] in 2015 described an adherence of 35.5% to geriatric interventions, and Kenis et al. [14] in 2018 adherence of over 40% in the most important domains. These three studies were conducted by the same team and suggest that the increased rate of implementation is obviously due to the learning curve. To our knowledge, there are no studies exploring the phenotype of patients who need more geriatric interventions. Screening patients who may need follow-up to check for intervention is new. 

### 3.2. Influence of the CGA on Treatment Changes

This study confirms the influence of the CGA on 22% of therapeutic decisions. This rate is very similar to that found in other studies. Kenis et al. [12] found 25%, Caillet et al. [17], 20%. Feasibility of treatment and guided geriatric interventions seem to have different mechanisms, but they are probably two sides of the same coin. Changes in treatment plan according to the CGA certainly influence the level and type of guided interventions. Furthermore, geriatricians implement different types of interventions depending on the type of treatment (chemotherapy, surgery, palliative care, etc.), and this process could probably lead to support deficits in areas of geriatric assessment and help in the treatment feasibility. These considerations need to be confirmed in further prospective studies in the PACA EST cohort. 

### 3.3. Factors Associated with an Increase in the Need for Intervention

In multivariate analyses, to be malnourished or at risk of being malnourished regarding the MNA, a PS >2, a dependence on IADL, a positive screening for depression regarding the GDS, cognitive disorders regarding an MMSE <24, and a positive screening regarding a G8 ≤14 are independent risk factors associated with an increased requirement of geriatric interventions. 

These are well-known factors for a worse outcome in geriatric oncology. Regarding the PS, there are several studies showing a prediction of an increased risk of death in this population [18,19,20]. In addition, there is abundant literature showing that the nutritional status and the MNA predict outcomes such as early death, early discontinuation of chemotherapy, poor tolerance of chemotherapy, and an increased risk of morbidities [21,22,23,24]. Mood disorders and cognitive impairment have also been explored in other studies showing early functional decline on chemotherapy and decreased survival [25,26,27]. Moreover, dependence on IADL was associated with increased mortality, morbidity, hospitalization, and functional decline [6]. These data invite us to take into account these factors to optimize the management and the follow up of elderly patients. However, we do not yet have robust data (randomized studies) in geriatric oncology supporting the fact that interventions improve the outcome, but some studies are underway to analyze the impact of a multimodal approach [6]. For example, the PREPARE study in France plans an interventional multimodal approach using “case management.” We hope that this study by Soubeyran et al. will supply abundant information and evidence in favor of guided interventions. (PREPARE, ClinicalTrials.gov Identifier: NCT02704832) [28].

### 3.4. A Call for Co-Management

This study shows that elderly patients require different interventions with various health partners. Creating individualized “care” and “take care” plans require a strong partnership between a network including geriatricians, oncologists, specialized doctors, but also nurses, psychologists, dieticians, social workers, physiotherapists, and many other actors. The task of the UCOG teams in France is to coordinate and streamline patient care, offering easy lines of communication and shortening referral times. In the domain of co-management, perspectives in geriatric oncology from innovative professions coordinating the development of specific e-health tools are plentiful [29]. 

### 3.5. How This Model Could Add Value in Clinical Practice?

The utility of the multivariable model lies in the determination of independent factors strongly associated with the establishment of more geriatric interventions. This model underlines some domains of the GA that can be assessed by other health partners, such as dieticians, nutritionists, psychologists, or psychiatrists, even if there is no geriatrician on the team. A nurse could coordinate and educate patients who present these factors and link them to general practitioners that can provide them with simple interventions such as nutritional support, advice, physiotherapy, and so on.

### 3.6. Improving Together Prediction and Outcome

Clinicians and researchers are working together to elaborate scoring systems or factors aimed at improving the outcome and patient care of older adults with cancer. The PACA EST cohort is a prospective and multicentric cohort (n > 3800) created to better understand elderly cancer patients. Prospective and systematic follow-up improve substantially the quality of care and connect general practitioners and heath partners. The UCOG PACA EST team joined the Hospital Federation University in research into OncoAge so that research and care become a continuum in the future. Subsequent studies will focus on adherence and on the impact of geriatric interventions in this cohort, but also on the barriers and difficulties of implementation. An OncoAge work package aims to improve and analyze a lifestyle plan. Educating patients and caregivers about the options of care, including guided geriatric interventions, is crucial.

### 3.7. Strengths and Limitations

The strengths of this study lie in the large cohort studied, which enrolled a “true life population” with a mean age of 82 years old. A complete and standardize CGA was performed at baseline. Interventions were guided by the deficits observed in the CGA. However, this study did not analyze the level of intervention and adherence at follow-up. A pilot study conducted in the PACA EST cohort (n = 50) had shown that, after one month, the adherence to interventions ranged from 73 to 89 % depending on the domains. This is probably because geriatricians in the PACA EST cohort implemented the interventions at baseline and did not propose recommendations only. 

## 4. Materials and Methods

### 4.1. Patient Population

Between April 2012 and May 2018, 3530 consecutive patients were enrolled in the PACA EST cohort. The UCOG PACA EST cohort is an observational, multicentric cohort (five centers in Southeast France: a teaching hospital, a specialized cancer center, and three cancer clinics). Three thousand one hundred and forty patients with various types of solid cancers at any stage and aged older than 70 years old (no upper limit) were enrolled in this study at the time of diagnosis and before the final therapeutic decision. Patients could be outpatients or hospitalized. Patients were referred by more than 60 practitioners (oncologists, surgeons, and radiotherapists) to the UCOG PACA EST team (Geriatric Oncology Coordination Unit) for a CGA before a final therapeutic decision. 

### 4.2. Ethics

At the first visit of inclusion, patients gave informed consent and were registered at baseline in compliance with the French database and privacy law (CNIL, Commission Nationale de l’Informatique et Liberté, registration number CILS: 188). This study was approved by an ethics committee (Espace Ethique Gériatrique Report 04-2012).

### 4.3. Study Methods

#### 4.3.1. CGA and Data Collected at Baseline

Four geriatricians received the same training at baseline and performed a standardized comprehensive geriatric baseline assessment as described in Table 1. The CGA included cognitive function screening using the Mini-Mental Test (MMSE) [30], an autonomy assessment using Activity in Daily Living (ADL) [31] and Instrumental Activity of Daily Living (IADL) [32,33], a nutritional status assessment using the Mini Nutritional Assessment [34], a gait assessment using gait speed [35], and the one leg balance test, screening for depression using the Geriatric Depression Scale 15 for patients with an MMSE score higher than 15 (GDS) [36], and acomorbidities assessment using the Charlson Index [37]. Prediction of early death was assessed with the Nice Cancer Aging Survival Score (NCASS) [21], and mortality at 4 years was assessed with the Lee score [38]. Demographic data and the perception of isolation and being homebound were also determined (homebound was defined in the study as going out of home with or without assistance only for important activities, e.g., a medical visit). Finally, the Balducci score was assessed [39]. The validated cut-offs for scales are specified in Table 2. In addition, data on guided geriatric interventions, on oncologic treatments proposed by oncologists, and on tumor type and tumor stage were collected during follow-up. Geriatric interventions were defined by interventions implemented by a geriatrician at baseline in 12 domains (nutrition, psychological care, specialized pain management, prevention of delirium, comorbidities management, nursing interventions, social worker interventions, treatment modification for optimization, adjustment medication for iatrogenic disorders, physiotherapy, caregiver care, and care pathway modification). Geriatric interventions are standardized (based on guidelines when available) and individualized (focused on specific deficits). Some interventions as caregiver care or social interventions are based on experience (no guidelines available). Geriatric interventions are described in Table 6. Within a month, geriatricians who included patients in the cohort received a specific training on the CGA and on guided interventions. They received a prescription book with standardized recommendations. 

#### 4.3.2. Statistics

The primary aim of the study was to analyze the association between geriatric and oncologic factors and the need to implement more than three geriatric interventions (the median number of guided interventions in the whole cohort) using a logistic regression model in a univariate analysis. A multivariable analysis was performed with all geriatric and oncologic items that reached a significant level of *p* < 0.05. Regarding frailty levels and interventions, we compared the medium number of geriatric interventions in the three groups according to the Balducci classification by using an ANOVA test and the Bonferroni adjusted *p*-value.

## 5. Conclusions

Nutritional status, a PS >2, a dependence on IADL, a positive screening for depression, cognitive impairment, and a G8 ≤14 were independent risk factors associated with more geriatric interventions. Factors associated with more geriatric interventions could assist practitioners in selecting patients for specific geriatric follow-up. Further studies on the PACA EST cohort will focus on the level of intervention and adherence at follow-up. Research needs to not only focus on interventions but also on the quality of the implementation according to the guidelines. Standardization of the interventions is an important task, and research is underway in a number of studies being performed in the world [11]. 

## Figures and Tables

**Table 1 cancers-11-00192-t001:** Demographic and tumor characteristics.

Demographic and Tumor Characteristics	n = 3140	%
**Age, years**		
Median 81.9 Range (70–102)		
<80	1286	41
80–85	978	31.1
>85	876	27.9
**Gender**		
Male	1395	44.4
**Cancer Site**		
Breast	548	17.5
Colorectal	525	16.7
Lung	356	11.3
Cholangiocarcinoma/pancreatic	281	8.9
Gynecological	226	7.2
Dermatologic	246	7.8
Bladder	219	7
Upper digestive	198	6.3
Head and neck	176	5.6
Prostatic	157	5
Kidney	94	3
Hepatocarcinoma	73	2.3
Other	41	1.4
**Stage IV**	1028	32.9
**ECOG-PS**		
0	260	8.3
1	966	30.8
2	855	27.2
3	807	25.7
4	226	7.2
>2	1033	32.9
Missing	26	0.8

PS: performance status.

**Table 2 cancers-11-00192-t002:** Comprehensive geriatric assessment (CGA) at baseline.

Comprehensive Geriatric Assessment Heading Title	n = 3140	%
Activity of Daily Living (ADL)		
≥5.5	1528	48.6
Missing	7	0.2
Instrumental Activity of Daily Living (IADL)		
>0	1885	60
Missing	8	0.3
Speed Gait		
<0.8 m/s	1482	47.2
Missing	5	0.2
One leg stand		
<5 s	2232	71.2
Missing	8	0.3
Isolation	242	7.7
Missing	6	0.2
Home confinement	896	28,6
Missing	4	0.1
Balducci Score		
1	146	4.6
2	1568	49.9
3	1426	45,4
Missing	0	
MNA		
>23.5	1030	32.8
17–23.5	1500	47.8
<17	502	16
Missing	108	3.4
MMSE		
≤24	1230	39.2
Missing	104	3.3
GDS		
<5	1912	69.9
Missing	249	7.9
G8 > 14	424	13.5
Missing	68	2.2
Lee Score		
0–5	52	1.7
0–9	763	24.3
0–13	1083	34.5
>14	1210	38
Missing	32	1
Ponderated Charlson		
<5	277	8.9
Missing	26	0.8
NCASS		
0–6	1592	50.7
7 to 9	762	24.2
8 to 9	490	15.6
11	138	4.5
Missing	158	5

ADL: Activity Daily Living; IADL: Instrumental Activity Daily Living; MNA: Mini Nutritional Assessment; GDS: Geriatric Depression Scale; MMSE: Mini Mental State Evaluation; NCCAS: Nice Cancer Aging Survival Score.

**Table 3 cancers-11-00192-t003:** Geriatric guided interventions.

Geriatric Interventions	n = 8819	%
Nutritional care	2231	71.1
Physiotherapist intervention	1462	46.6
Delirium prevention	599	19.1
Social worker interventions	733	23.3
Psychological/Psychiatric care	510	16.2
Treatment modification for optimization	667	21.2
Adjustment medication for iatrogenic disorders	351	11.2
Comorbidity management	970	30.9
Nursing interventions	580	18.5
Specialized pain management	96	3.1
Caregiver care	355	11.3
Care pathway modification	265	8.4

**Table 4 cancers-11-00192-t004:** Univariate significant factors of an increased need of geriatric interventions.

Geriatric Interventions (GI)	3 GI n = 999	%	<3 GI n = 2137	%	*p* value
Dependence on ADL	598	59.9	935	43.8	*p* < 0.0001
Dependence on IADL	743	74.4	1147	53.7	*p* < 0.0001
Speed gait					
<0.8m/s	572	57.4	908	42.5	*p* < 0.0001
Isolation	103	10.3	140	6.6	*p* < 0.0001
Delirium	79	7.9	97	4.5	*p* < 0.0001
Home Confinement	436	43.6	461	21.6	*p* < 0.0001
MNA score					
17–23.5	531	55.3	968	46.8	*p* < 0.0001
<17	256	27.0	242	11.7	*p* < 0.0001
MMSE					
≤24	508	52.6	721	34.9	*p* < 0.0001
GDS					
≥5	427	46.9	550	27.8	*p* < 0.0001
G8 score					
>14	928	94.6	1716	82.2	*p* < 0.0001
Charlson score					
≥6	931	93.8	1904	89.9	*p* < 0.0001
Stage IV					
	361	36.4	666	31.3	*p* = 0.005
Performance status					
>2	472	47.4	562	26.6	*p* < 0.0001

ADL: Activity Daily Living; IADL: Instrumental Activity Daily Living; MNA: Mini Nutritional Assessment, GDS: Geriatric Depression Scale; MMSE: Mini Mental State Evaluation.

**Table 5 cancers-11-00192-t005:** Independent factors associated with an increased need of geriatric interventions.

Factors	*p*	OR	95%CI
G8 ≤ 14	0.023	1.5	(1.1–2.1)
Dependence on IADL	0.013	1.3	(1.1–1.6)
MNA score			
>23.5			Reference
17–23.5	<0.0001	1.9	(1.5–2.4)
<17	<0.0001	3.1	(2.2–4.3)
GDS ≥ 5	<0.0001	1.5	(1.2–1.8)
MMS ≤ 24	0.009	1.3	(1.1–1.5)
PS > 2	*p* = 0.003	1.4	(1.1–1.8)

MNA: Mini Nutritional Assessment; PS: Performance Status; MMS: Mini Mental State; GDS: Geriatric Depression Scale. Adjusted to metastatic status, age, comorbidity index, and ADL.

**Table 6 cancers-11-00192-t006:** Guided geriatric intervention description.

Interventions	Description
**Nutritional Care**	Nutritional Advice Nutritional supplements Artificial nutrition **Based on guidelines [40,41], standardized prescription**
**Physiotherapist Interventions**	Balance Strength Pain management Recommendations of walking aids Coordination Promotion of physical activity **Based on patient deficits, standardized prescription (list)**
**Delirium Prevention**	Checklist for patient, caregiver and medical team: advice, recommendations for prescription for surgical team. **Based on guidelines [42], standardized check list**
**Social Worker Interventions**	Prevention, In home health services, housing, social inclusion, financial accommodations, legal action, end of life services, institutional placement, nutrition accommodations. **Based on social worker and geriatrician experiences.**
**Psychological/Psychiatric Care**	Consultation with psychologist or psychiatrist **Duration and methods based on patient needs and practitioners experience**
**Treatment Modification for Optimization**	Medical treatment assessment, optimization of treatment **Based on geriatrician experience**
**Adjustment Medication for Iatrogenic Disorders**	Inappropriate medication assessment. **Based on geriatrician experience**
**Comorbidity Management**	Advice, treatment modification, referral to others clinicians or paramedical, medical checkup **Based on geriatrician experience**
**Nursing Interventions**	
**Specialized Pain Management**	Drug or non-drug therapy, referral to specific pain management **Based on guidelines [43]**
**Caregiver Care**	Counselling, training courses, social supports, medical supports, psychological care, assistance bureaucracies, advocacy, crisis interventions **Based on geriatrician and social worker experiences**
**Care Pathway Modification**	Identification of appropriate resources, coordination of the care process, coordination of admission in acute care unit rehabilitation unit (rehabilitation/prehabilitation), long stay hospitalization, referral to a one-day hospital, integration on specific organization (palliative care, home care hospitalization) **Based on geriatrician and social worker experiences**

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
