# Peer review of "The Desire to Better Understand Older Adults with Solid Tumors to Improve Management: Assessment and Guided Interventions—The French PACA EST Cohort Experience"

_cancers, 2019, doi:10.3390/cancers11020192_

Round 1
Reviewer 1 Report
This reviewer applauds the efforts in France to appropriately care for older patients with advanced cancer using a systematic approach. The strength of this analysis is the evaluation of interventions by frailty status which has minimally been reported in the literature. Having said this, there are improvements that could be made:
the introduction could provide more of a background that includes definition of CGA, its importance, and a summary of where the research is in the field (that CGA has been shown to predict outcomes and guide both decisions and interventions). SIOG and ASCO guidelines sould be referenced.
the introduction could then better highlight the gaps in knowledge--how best to employ/allocate teams to conduct geriatric interventions
Not clear why methods are after results unless this is the journal's preference
4. the prevalence of treatment changes due to CGA is discussed, but not much information is provided. Has this been published previously? If not understanding how GA guided decisions would be helpful.
5. The methods state that geriatricians received similar training to conduct CGA. Did this include training on providing the interventions? How were the interventions standardized? How were they developed? Or were these provided based on geriatricians' clinical expertise?
6. While there is a table on interventions, further description of what interventions were provided for each category would be helpful. For example, more information on what was provided for caregiver interventions would help strengthen the manuscript.
7. While the number of interventions differed by underlying frailty status, did the kind of interventions differ also?
8. Its not clear how the multivariate analysis adds value. More explanation would be helpful. Because the CGA measures triggered or guided interventions, wouldn't it make sense that the abnormal measures are associated with interventions? How does this add value?
9. More discussion on generalizeability of results would strengthen the manuscript. How could others follow this model of care? Especially in areas where geriatricians are not accessable. Also some more discussion on adherence would strengthen the manuscript. The interventions were recommended but were they also adhered to?
Author Response
REVIEWER 1
Dear Xinya Huang, Dear reviewers
First of all, we would like to thank the Reviewers for their high quality and constructive reviews of our manuscript (Entitled: The Desire to Better Understand Older Adults with Solid Tumors to Improve Management: Assessment and Guided Interventions. The French PACA EST Cohort Experience) as well as the Editor for his/her careful reading. In the revised version of the manuscript, we addressed all comments raised by the Reviewers and you will found our responses point by point. We agree with most reviewers' comments and made changes. These comments led to improve and clarify the manuscript and its scientific message for the readers of Cancers (Basel).
We hope that our revised manuscript will reach now the standards of Cancers (Basel) for publication and we thank you for your consideration concerning our present work
Sincerely yours,
Responses to reviewer 1
We would like the reviewer1 for the helpful comments and we have revised our manuscript accordingly to the different points raised
POINT 1 « the introduction could provide more of a background that includes definition of CGA, its importance, and a summary of where the research is in the field (that CGA has been shown to predict outcomes and guide both decisions and interventions). SIOG and ASCO guidelines should be referenced. the introduction could then better highlight the gaps in knowledge--how best to employ/allocate teams to conduct geriatric interventions »
We added in the introduction the definition of the CGA (Rubenstein MD): Comprehensive geriatric assessment as defined by Rubenstein is a “multidimensional interdisciplinary diagnostic process focuses on determining a frail elderly person’s medical, psychological and functional capability in order to develop coordinated and integrated plan “
We underlined now in the introduction the ASCO and SIOG guidelines regarding the CGA with new references. We also highlighted how the CGA can predict patient ‘outcomes (toxicities and life expectancy) and we added some references. Finally, we also proposed to precise the main purpose of this study which was to screen elderly patients who really need to perform repeated CGA during their follow up.
Changes in the paper regarding point 1:
In clinical routine, the International Society of Geriatric Oncology (SIOG) and the American Society of Clinical Oncology (ASCO) recommend performing a Comprehensive Geriatric Assessment (CGA) due to the substantial heterogeneity among elderly patients [5,6]. Comprehensive Geriatric Assessment (CGA) as defined by Rubenstein is a “multidimensional interdisciplinary diagnostic process focused on determining a frail elderly person’s medical, psychological and functional capability in order to develop coordinated and integrated plan for treatment” [7]. The CGA is time consuming but specific tools for frailty screening are available to detect patients who really need to perform a complete CGA [8,9,10]. During the past decade, the partnership between geriatricians and oncologists has improved patient care by profiling the level of patient frailty with this process. Therefore, the CGA has been shown to predict outcomes (chemotherapy toxicity, life expectancy), to help make therapeutic decisions, but also provide with the best interventions [11,12]. Previous studies have described adherence of geriatric assessment recommendations [13,14], and also guidelines for practical assessment and management of older patients receiving chemotherapy [6]. Now, the challenge in geriatric oncology is to screen patients who require geriatric follow-up with specific guided interventions. The main question is to determine which patients need to have repeated geriatric assessment in the follow-up. So, the goal of this study was to analyze a large cohort of patients with solid tumors, for factors that provide a profile of the phenotype of patients who need more geriatric interventions and therefore specific follow-up.
References (Point 1)
Wildiers, H; Heeren, P; Puts, M; Topinkova, E; Janssen-Heijnen, ML; Extermann, M;Falandry,C,;Artz, A; Brain, E; Colloca, G;Flamaing, J; Karnakis, T; Kenis, C; Audisio, RA;Mohile, S; Repetto, L; Van Leeuwen, B; Milisen, K; Hurria, A. International Society of Geriatric Oncology consensus on geriatric assessment in older patients with cancer. J Clin Oncol. 2014,32,2595-603. DOI:10.1200/JCO.2013.54.8347
Mohile, SG; Dale, W; Somerfield, MR; Hurria A. Practical Assessment and Management of Vulnerabilities in Older Patients Receiving Chemotherapy: ASCO Guideline for GeriatricOncology Summary. J Oncol Pract, 2018,14,442-446. doi: 10.1200/JOP.18.00180.
Rubenstein, LZ ; Stuck, AE ; Siu, AL ; Wieland, D. Impact of geriatric evaluation and
management programs on defined outcomes: overview of the evidence.
JAGS. 1991,39,8–16
Hurria, A ; Togawa, K ; Mohile, SG ; Owusu, C ; Klepin, HD ;Gross, CP ; Lichtman, SM ;Gajra, A ; Bhatia, S ; Katheria, V ; Klapper, S ; Hansen, K ; Ramani, R ; Lachs, M ; Wong, FL ;Tew, WP. Predicting chemotherapy toxicity in older adults with cancer: a prospective multicenter study. J Clin Oncol. 2011,29,3457-65. doi:10.1200/JCO.2011.34.7625.
Extermann, M ; Boler, I; Reich, RR ; Lyman, GH ; Brown, RH ; DeFelice, J ;Levine, RM ; Lubiner, ET ; Reyes, P ; Schreiber,FJ 3rd ; Balducci, L. Predicting the risk of chemotherapy toxicity in older patients: the Chemotherapy Risk Assessment Scalefor High-Age Patients (CRASH) score. Cancer. 2012,118,3377-86. doi:10.1002/cncr.26646.
POINT 2 “Not clear why methods are after results unless this is the journal's preference”
It is the journal’s “preference”. (Research manuscripts sections: Introduction, results, discussion, Materials and Methods, Conclusions)
POINT 3 “the prevalence of treatment changes due to CGA is discussed, but not much information is provided. Has this been published previously? If not understanding how GA guided decisions would be helpful”
We would like to underline that in the present study, the changes in treatment were attributed to geriatric assessment alone if it was noted in the multidisciplinary meeting reports and/or in the oncologist’s medical files. It is essential to bear these considerations in mind because changes could be due to other factors: choice of patients or physicians, clinical deterioration or biological changes.
We published recently in the European Journal of Cancer (2018) similar results regarding the influence of the CGA on the treatment changes (on 1020 patients). As a geriatric team having a strong partnership with oncologists, surgeons and radiotherapists, we observed that treatment (plan and feasibility) and guided interventions are linked in clinical routine practice. This is the reason why we decided to emphasize the prevalence of treatment changes due to CGA.
Changes in the paper regarding point 2
We added at the beginning of the discussion a few sentences:
3.2 Influence of the CGA on treatment changes
This study confirms the influence of the CGA on 22% of therapeutic decisions. This rate is very similar to that found in other studies. Kenis et al [12], found 25% and Caillet et al [17], 20%. Feasibility of treatment and guided geriatric interventions seems to have different mechanisms, but they are probably two sides of the same coin. Changes in treatment plan according to the CGA certainly influence the level and type of guided interventions. Furthermore, Geriatricians implement different type of interventions according to the type of treatment (chemotherapy, surgery, palliative care…) and this process could probably lead to support deficits in areas of geriatric assessment and help in the treatment feasibility. These considerations need to be confirmed in further prospective studies in the PACA EST cohort.
Reference added for the point 2:
Caillet, P; Canoui-Poitrine, F; Vouriot, J; Berle, M; Reinald, N; Krypciak, S;Bastuji-Garin, S; Culine, S; Paillaud ,E. Comprehensive geriatric assessment in the decision-making process in elderly patients with cancer: ELCAPA study. J Clin Oncol. 2011, 29, 3636-42. doi: 10.1200/JCO.2010.31.0664.POINT 4 “The methods state that geriatricians received similar training to conduct CGA. Did this include training on providing the interventions? How were the interventions standardized? How were they developed? Or were these provided based on geriatricians' clinical expertise?” POINT 5 “While there is a table on interventions, further description of what interventions were provided for each category would be helpful. For example, more information on what was provided for caregiver interventions would help strengthen the manuscript.”
Four geriatricians received the same exact training. We provided for each geriatrician who included patients in the cohort, a documentation with standardized prescriptions. Within one month, they received a specific training on the CGA and on guided interventions.
Some interventions are well standardized (nutrition according to French guidelines, as an example) but others are more focused on deficits and based on clinical expertise (duration of psychological care for example, or caregiver care). This study shows that elderly patients require different interventions with different partners. Some of them proposed an individualized care plan (physiotherapist, psychologist for example). The task of the UCOG PACA EST team is to coordinate all these health partners and to assess these actions. Concerning caregivers, interventions offer a large variety of options (psychological care, social intervention, training, medical care…)
Changes in the paper regarding point 4 and 5:
We added a sentence regarding the training in the study methods.
Geriatric interventions are standardized (based on guidelines when available) and individualized (focused on specific deficits). Some interventions as caregiver care or social interventions are based on experience (no guidelines available). Geriatric interventions are described in Table 6. Within 1 month, geriatricians who included patients in the cohort received a specific training on the CGA and on guided interventions. They received a prescription book with standardized recommendations.
We added the table 6
Table 6
INTERVENTIONS | Description | |
Nutritional care | Nutritional Advices Nutritional supplements Artificial nutrition Based on guidelines [40,41], Standardized prescription | |
Physiotherapist interventions | Balance Strength Pain management Recommendations of walking aids Coordination Promotion of physical activity Based on patient deficits, Standardized prescription.(list) | |
Delirium prevention | Check list for patient, caregiver and medical team: advices, recommendations for prescription for surgical team. Based on guidelines [42], Standardized check list | |
Social worker interventions | Prevention, In home health services, housing, social inclusion, financial accommodations, legal action, end of life services, institutional placement, nutrition accommodations. Based on social worker and geriatrician experiences. | |
Psychological/Psychiatric care | Consultation with psychologist or psychiatrist Duration and methods are based on patient needs and practitioners experience | |
Treatment modification for optimization | Medical treatment assessment, optimization of treatment Based on geriatrician experience | |
Adjustment medication for iatrogenic disorders | Inappropriate medication assessment. Based on geriatrician experience | |
Comorbidity management | Advices, Treatment modification, referral to others clinicians or paramedical, medical checkup Based on geriatrician experience | |
Nursing interventions | ||
Specialized pain management | Drug or non drug therapy, referral to specific pain management Based on guidelines [43] | |
Caregiver care | Counselling, training courses, social supports, medical supports, psychological care, assistance bureaucracies, advocacy, crisis interventions Based on geriatrician and social worker experiences | |
Care pathway modification | Identification of appropriate resources, coordination of the care process, coordination of admission in acute care unit rehabilitation unit (rehabilitation/prehabilitation), long stay hospitalization, referral to a one-day hospital, integration on specific organization (palliative care, home care hospitalization) Based on geriatrician and social worker experiences |
Haute Autorité de Santé. Nutritional Support Strategy for Protein-Energy. Malnutrition in the Elderly. 2007. http://www.hassante.fr/portail/plugins/ModuleXitiKLEE/types/FileDocument/doXiti.jsp?id=c_630900. Accessed January 29, 2014 http://www.hassante.fr/portail/plugins/ModuleXitiKLEE/types/FileDocument/
French Speaking Society of Clinical Nutrition and Metabolism (SFNEP). Clinical nutrition guidelines of the French Speaking Society of Clinical Nutrition and Metabolism (SFNEP): Summary of recommendations for adults undergoing non-surgical anticancer treatment. Dig Liver Dis. 2014,46,667-74. doi: 10.1016/j.dld.2014.01.160.
Bush, SH; Lawlor, PG; Ryan, K; Centeno, C; Lucchesi, M; Kanji, S;Siddiqi, N; Morandi,
A;Davis, DHJ; Laurent, M;Schofield, N; Barallat, E; Ripamonti, CI. ESMO Guidelines
Committee. Delirium in adult cancer patients: ESMO Clinical Practice Guidelines.
Ann Oncol. 2018 Oct 1;29(Supplement_4):iv143-iv165. doi: 10.1093/annonc/mdy147.
Evaluation et suivi de la douleur chronique chez l’adulte en médecine ambulatoire, ANAES : http://www.has- sante.fr/portail/jcms/c_540915/evaluation-et-suivi-de-la-douleur-chronique-chez-l-adulte-en-medecine-ambulatoire
POINT 6 “While the number of interventions differed by underlying frailty status, did the kind of interventions differ also?”
We also analyze the type of interventions according to the frailty status. The kind of interventions without surprise is different. For example, fit patients had few interventions such as modification of care pathway, caregiver care or pain management. We have other data on this analysis which ultimately deserves a new paper. Because of embargo policy regarding this new submission we can’t integrate these data on the present manuscript.
POINT 7 “Its not clear how the multivariate analysis adds value. More explanation would be helpful. Because the CGA measures triggered or guided interventions, wouldn't it make sense that the abnormal measures are associated with interventions? How does this add value? ”POINT 8 “More discussion on generalizability of results would strengthen the manuscript. How could others follow this model of care? Especially in areas where geriatricians are not accessible. Also some more discussion on adherence would strengthen the manuscript. The interventions were recommended but were they also adhered to?”
We totally agree with the fact that there is a link between domains of the CGA and guided interventions. The utility of the multivariate model lies in the determination of independent factors (scales, tools) strongly associated with the establishment of more geriatric interventions (>3). The aim of the present work was not to determine if there is a link between CGA and interventions, but we wanted to know if the influence of some factors was important and led to more interventions. On one hand, for example, lower gait speed will probably have triggered the physiotherapist intervention and sometimes other interventions but we did not found this assessment in the multivariate model. On the other hand, the G8 and the MNA are independently associated with more interventions, probably because geriatrician concerning nutritional domain propose at least physiotherapy, nutritional care and modification of medications, so the nutritional status can also reflect other serious conditions which could lead to more interventions (metastatic cancer and/or severe depression for example). Lower gait speed in some cases is only associated to some orthopedic issues.
As emphasized by the reviewer, the second reason why this model has an add value is that it is very helpful in areas where geriatricians are not accessible or available. In that case, it’s necessary to provide simple models and factors that could alarm medical teams on the need to make a specific assessment and follow-up. So how these teams could use these results in clinical practice? This model underlines some domains of the GA which can be assessed also by other health partners such as dieticians, or nutritionists (MNA), the G8 could be easily made by a nurse or by an oncologist. Patients who seem to have cognitive or mood disorders could be referred to psychologist or psychiatrics even if there is no geriatrician in the team. At least, a nurse could coordinate and educate patients who present these factors and make the link with the GP to provide them simple interventions such as nutritional support, advices, physiotherapy, etc.…
We agree with the reviewer when his/her states that it is crucial to discuss on adherence. Our study did not analyze the level of intervention and the adherence during the follow up. But a pilot study, on a set of patients (n=50) had shown that after one month the adherence to interventions ranges from 73% to 89 % depending on the domains. Probably because in the PACA EST COHORT, geriatricians have implemented the interventions at baseline and did not propose only recommendations.
We added in the discussion:
How this model could add value in clinical practice?
The utility of the multivariate model lies in the determination of independent factors strongly associated with the establishment of more geriatric interventions. This model underlines some domains of the GA which can be assessed also by others health partners such as dietician, nutritionist, psychologist or psychiatrics even if there is no geriatrician in the team. At least, a nurse could coordinate and educate patients who present these factors and make the link with the general practitioners to provide them simple interventions such as nutritional support, advices, physiotherapy…
And in the last paragraph:
A pilot study in the PACA EST Cohort (n=50) had shown that after one month the adherence to interventions ranges from 73% to 89 % depending on the domains. Probably because geriatricians in the PACA EST cohort, have implemented the interventions at baseline and did not propose only recommendations.

Reviewer 2 Report
The manuscript of “The desire to better understand older adults with solid tumors to improve management: assessment and guided interventions. The French PACA EST cohort experience” descripts the elderly cancer patients’ patterns of geriatric assessment (GI) and factors associated with GI. This study provided important information on how to care for geriatric cancer patients, but there are some issues with this paper that should be addressed prior to considering publication.
Many of the abbreviations such as PACA EST, MNA, PS, IADL etc., in the abstract session that should be spell out. Please make sure all abbreviations in tables were spelt out such as ECOG-PS in Table 1.
The study variables for the geriatric assessment such as activity of daily living with the numeric value of <5.5 or >=5.5 can be defined as high or low, or in a meaningful way to help the readers easily understand a patient’s frailty or functionality status. Also, use the cutoff more consistent such as the age group should be cutoff at <80, 80-85, and >85 years, please indicate which group including the 80 or 85 years old.
In the introduction session, use two or three sentences to describe the main purpose of this study such as describe the use of geriatric assessment of elderly cancer patients and factors associated with patients who used more geriatric follow-ups.
The logistic regression model should be called “multivariable” instead of multivariate since there was only one outcome variable in that model.
Author Response
REVIEWER 1
Dear Xinya Huang, Dear reviewers
First of all, we would like to thank the Reviewers for their high quality and constructive reviews of our manuscript (Entitled: The Desire to Better Understand Older Adults with Solid Tumors to Improve Management: Assessment and Guided Interventions. The French PACA EST Cohort Experience) as well as the Editor for his/her careful reading. In the revised version of the manuscript, we addressed all comments raised by the Reviewers and you will found our responses point by point. We agree with most reviewers' comments and made changes. These comments led to improve and clarify the manuscript and its scientific message for the readers of Cancers (Basel).
We hope that our revised manuscript will reach now the standards of Cancers (Basel) for publication and we thank you for your consideration concerning our present work
Sincerely yours,
Responses to reviewer 2
We would like to thank the reviewer 2 for the helpful comments and we have revised our manuscript accordingly to the different points raised.
POINT 1: “Many of the abbreviations such as PACA EST, MNA, PS, IADL etc., in the abstract session that should be spell out. Please make sure all abbreviations in tables were spelt out such as ECOG-PS in Table 1.”
We checked now all the abbreviations in the paper. Please note that PACA EST is not an abbreviation, it is the name of the PACA (“Provence Alpes Côte d’Azur”) area in the South East of France and the name of our medical unit.
POINT 2 : “The study variables for the geriatric assessment such as activity of daily living with the numeric value of <5.5 or >=5.5 can be defined as high or low, or in a meaningful way to help the readers easily understand a patient’s frailty or functionality status. Also, use the cutoff more consistent such as the age group should be cutoff at <80, 80-85, and >85 years, please indicate which group including the 80 or 85 years old.”
It is a very relevant comment. We checked the numeric values and made changes in the tables accordingly.
POINT 3: “In the introduction session, use two or three sentences to describe the main purpose of this study such as describe the use of geriatric assessment of elderly cancer patients and factors associated with patients who used more geriatric follow-ups.”
Regarding this comment, we modified the introduction to clarify the main purpose of this study:
We added:
In clinical routine, the International Society of Geriatric Oncology (SIOG) and the American Society of Clinical Oncology (ASCO) recommend performing a Comprehensive Geriatric Assessment (CGA) due to the substantial heterogeneity among elderly patients [5,6]. Comprehensive Geriatric Assessment (CGA) as defined by Rubenstein is a “multidimensional interdisciplinary diagnostic process focused on determining a frail elderly person’s medical, psychological and functional capability in order to develop coordinated and integrated plan for treatment” [7]. The CGA is time consuming but specific tools for frailty screening are available to detect patients who really need to perform a complete CGA [8,9,10]. During the past decade, the partnership between geriatricians and oncologists has improved patient care by profiling the level of patient frailty with this process. Therefore, the CGA has been shown to predict outcomes (chemotherapy toxicity, life expectancy), to help make therapeutic decisions, but also provide with the best interventions [11,12]. Previous studies have described adherence of geriatric assessment recommendations [13,14], and also guidelines for practical assessment and management of older patients receiving chemotherapy [6]. Now, the challenge in geriatric oncology is to screen patients who require geriatric follow-up with specific guided interventions. The main purpose is to determine which patients need to have repeated geriatric assessment in the follow-up. So, the goal of this study was to analyze a large cohort of patients with solid tumors, for factors that provide a profile of the phenotype of patients who need more geriatric interventions and therefore specific follow-up.
POINT 4: “The logistic regression model should be called “multivariable” instead of multivariate since there was only one outcome variable in that model.”
We changed multivariate by multivariable.

Round 2
Reviewer 1 Report
All comments addressed
Reviewer 2 Report
In abstract session, please spell out what is: MNA, PS, IADL, GDS, MMSE, and G8.